# Serum Concentrations of the Endocannabinoid, 2-Arachidonoylglycerol, in the Peri-Trauma Period Are Positively Associated with Chronic Pain Months Later

**DOI:** 10.3390/biomedicines10071599

**Published:** 2022-07-05

**Authors:** Colleen M. Trevino, Cecilia J. Hillard, Aniko Szabo, Terri A. deRoon-Cassini

**Affiliations:** 1Division of Trauma and Acute Care Surgery, Department of Surgery, Medical College of Wisconsin, Milwaukee, WI 53226, USA; tcassini@mcw.edu; 2Department of Pharmacology and Toxicology, Neuroscience Research Center, Medical College of Wisconsin, Milwaukee, WI 53226, USA; chillard@mcw.edu; 3Department of Biostatistics, Medical College of Wisconsin, Milwaukee, WI 53226, USA; aszabo@mcw.edu; 4Comprehensive Injury Center, Medical College of Wisconsin, Milwaukee, WI 53226, USA

**Keywords:** *N*-arachidonoylethanolamine, cortisol, chronic pain, injury

## Abstract

Endocannabinoid signaling and the hypothalamic-pituitary-adrenal axis are activated by trauma and both stress systems regulate the transition from acute to chronic pain. This study aimed to develop a model of relationships among circulating concentrations of cortisol and endocannabinoids (eCBs) immediately after traumatic injury and the presence of chronic pain months later. Pain scores and serum concentrations of eCBs and cortisol were measured during hospitalization and 5–10 months later in 147 traumatically injured individuals. Exploratory correlational analyses and path analysis were completed. The study sample was 50% Black and Latino and primarily male (69%); 34% percent endorsed a pain score of 4 or greater at follow-up and were considered to have chronic pain. Path analysis was used to model relationships among eCB, 2-arachidonolyglycerol (2-AG), cortisol, and pain, adjusting for sex and injury severity (ISS). Serum 2-AG concentrations at the time of injury were associated with chronic pain in 3 ways: a highly significant, independent positive predictor; a mediator of the effect of ISS, and through a positive relationship with cortisol concentrations. These data indicate that 2-AG concentrations at the time of an injury are positively associated with chronic pain and suggest excessive activation of endocannabinoid signaling contributes to risk for chronic pain.

## 1. Introduction

Acute pain is inevitable and important following injury as it protects the individual against further tissue damage. However, pain that persists after tissue injury has healed, so-called “chronic pain,” is not protective and has a significant, negative effect on the quality of life [1]. Chronic pain is defined as pain persisting after surgery or trauma for greater than three months [2]. Previous studies from our group and others have found that the incidence of chronic pain in traumatically injured patients can be as high as 70% and there is a strong correlation between pain severity and life interference [3,4,5]. Unlike acute pain, chronic pain is not primarily related to tissue injury [6] and traditional therapies for pain, including opioids, have poor efficacy in their treatment [7,8]. Not all injured individuals develop chronic pain, and few reliable or clinically significant biomarkers have been identified that predict the progression of acute to chronic pain in the traumatically injured population. Thus, this research could contribute to an improvement in our ability to predict who will develop chronic pain through the validation of biomarkers. A secondary long-term goal of this research project is to better understand the biological factors that contribute to the development of chronic pain, which could improve our ability to treat or prevent this transition.

Preclinical research suggests a role for the endocannabinoid signaling system (ECSS) in pain [9,10]. The endogenous ligands for cannabinoid (CB) receptors, called endocannabinoids (eCBs), are *N*-arachidonoylethanolamine (AEA or anandamide) and 2-arachidonoylglycerol (2-AG). Noxious stimuli and tissue injury increase eCB mobilization [11,12] and chronic pain can up-regulate the expression of CB1 [13] and CB2 [14] receptors. Although preclinical studies largely support the hypothesis that endogenous activation of CB receptors reduces acute pain [9,10], eCBs produced by intense nociceptive stimuli render nociceptive neurons in the spinal cord excitable by non-painful stimuli, suggesting that eCB/CB signaling can also promote pain sensitization and thus contribute to the development of chronic pain [15].

Multiple studies have examined the relationships between the concentrations of circulating eCBs and related lipids with the presence of chronic pain in humans. Circulating concentrations of 2-AG are higher compared to pain-free control groups in several types of chronic pain, including fibromyalgia, irritable bowel syndrome, and neuropathic pain [16,17,18]. Similarly, circulating AEA concentrations in individuals with fibromyalgia are three times higher than in matched controls [19]. Patients with complex regional pain syndrome (CRPS), a neuropathic pain syndrome precipitated by extremity injury, have significantly higher plasma concentrations of AEA than controls [20]. Women with endometriosis-associated pain, compared to those without endometriosis, exhibited elevated concentrations of both AEA and 2-AG [21]. Thus, while preclinical data indicate that eCB-mediated signaling reduces pain sensation at multiple sites within the neuronal pain circuit, the peripheral pool of eCBs is positively associated with chronic pain in humans. However, the presence of an association is not indicative of a causal relationship; it is possible that the high circulating eCBs are a physiological response to the stress of the pain experience.

Human studies also support the role of cortisol in developing chronic pain [22,23,24]. The imposition of acute pain (such as the cold pressor test) in healthy individuals elicits cortisol secretion, which is in accord with pain as a stressor [25]. On the other hand, inappropriate activation of the hypothalamic-pituitary-adrenal (HPA) axis is associated with the presence or development of chronic pain [26,27]. In particular, inappropriately low concentrations of cortisol have been associated with a diagnosis of chronic pain from fibromyalgia and low back pain [28,29].

Both the HPA axis [30] and the ECSS [31] are activated by exposure to physical and psychological stress, and these systems have diverging and converging effects on the stress response. HPA axis activation via actions of cortisol and the ECSS reduce pain and inflammation. In the brain, there are considerable data that cortisol increases the 2-AG synthesis and that CB1 receptor activation links brain cortisol to changes in synaptic activity. On the other hand, ECSS activation in the brain reduces HPA axis activation by stress and enhances recovery to baseline following stress. Importantly, both the ECSS and HPA axis are downregulated in situations of chronic stimulation, so excessive activation of either system can lead to loss of critical homeostatic processes.

This study aimed to develop a model of the relationships between circulating cortisol and eCBs, and the development of chronic pain after a traumatic injury. Our working hypothesis is that both the HPA axis and the ECSS are mobilized by the severe stress that accompanies traumatic injury and both contribute to reduced pain at that time point. However, we hypothesize that excessive concentrations of both at the time of injury will increase the risk for chronic pain development. The specific hypotheses for this study were: (1) circulating concentrations of the eCBs positively correlate with circulating cortisol concentrations at the time of injury, both reflecting the physical and psychological stress of the injury; (2) at the time of injury, pain measures negatively correlate with circulating eCB and cortisol concentrations due to the ability of both mediators to reduce pain; and (3) circulating eCBs concentrations at the time of hospitalization positively correlate with measures of pain months after the injury, possibly due to down-regulation of the ECSS.

## 2. Materials and Methods

### 2.1. Participants

The Institutional Review Board (IRB) at the Medical College of Wisconsin approved all study procedures (PRO00022827, approved on 3 May 2019) and participants were monetarily compensated for their time. Participants in this study were subjects of a prospective, exploratory, longitudinal cohort study entitled “Study on Trauma and Resilience (STAR).” We have previously reported some demographic and clinical data and the relationship between endocannabinoids and depression [32] and risk for post-traumatic stress disorder (PTSD) [33] in the same cohort. Two hundred eighty participants were recruited and consented at the time of injury in the parent study; 147 completed the follow-up pain assessments and blood draw and are included in these analyses.

### 2.2. Study Design

Individuals with any type of traumatic injury who were admitted to the inpatient trauma service at Froedtert Hospital, a level 1 American College of Surgery verified trauma center, were eligible for recruitment. Recruitment occurred over a 19-month period by daily review of the trauma division inpatient census for those who experienced a traumatic event; were at least 18 years of age, English speaking, and able to provide written informed consent within seven days of admission. Excluded were those who did not have appropriate cognitive capacity defined as Glasgow Coma Scale of 13 or less (e.g., moderate or severe TBI; obtained from chart review) and greater than 30 min of peritraumatic amnesia; were in police custody, or were having active psychotic or self-harm symptoms. Participants returned to the campus translational research unit 5–10 months (average 192 days, range 156–286) after their injury for a follow-up visit.

### 2.3. Measures

After providing informed consent and during hospitalization, study participants completed a series of questionnaires as a part of the parent study of trauma and resilience. Participant demographics, injury-related data, and a blood sample were obtained. The mean time of blood sampling was 1156 h, SD 1.6 h (approximately noon). The pain was assessed at the time of hospitalization via the numeric pain score (NPS), using a Likert scale with anchors at 0 (no pain) and 10 (worst pain imaginable) [34]. We utilized the injury severity score (ISS) as an anatomical measure of the severity of multiple physical traumatic injuries based upon the worst injury of six body systems [35]. Each system is scored from 1–6 depending on the level of severity and the sum of squares is taken from the three most injured systems. The highest score is 75 and denotes a non-survivable injury. The ISS for mild injury is 1–8, moderate injury is 9–15, severe injury is 16–24, and very severe is 25 and higher. The ISS was measured once at the time of hospitalization.

At the follow-up visit, blood was collected, and questionnaires were administered. The mean time of blood collection at follow-up was 1217 h, SD = 2.39 h. The Brief Pain Inventory (BPI) [36] was collected at the follow-up visit. The BPI measures both the intensity of the pain (sensory dimension) and interference of pain in the patient’s life (reactive dimension), with higher scores indicating greater pain intensity and interference, respectively [36]. Pain measures were asked within the context of the patient’s initial traumatic injuries. In our exploratory analyses, we used NPS of equal to or greater than 4 out of 10 to determine chronic pain since moderate and severe pain are associated with compromised physical functioning [37].

### 2.4. Study Procedures

Whole blood samples were drawn at hospitalization and the follow-up visit using serum collecting tubes (red-top tubes). After incubation at room temperature for 30–60 min, serum was harvested by centrifugation. Serum concentrations of the eCBs (2-AG and AEA) were measured in lipid extracts using isotope dilution and liquid chromatography-mass spectrometry to quantify daughter ions of AEA and 2-AG as described previously [38]. Concentrations of cortisol were measured in the same serum samples using radioimmunoassay (Cort-Cote 06B256440; MP Biomedical, San Diego, CA, USA). The sensitivity for cortisol assay was 57.5 pg/mL and no data lower than the minimum detection level were found. Based on the manufacturer’s reporting, intra-assay precision varies from 7.3–10.5, and inter-assay precision varies from 8.6–13.4 for high-to-low cortisol levels.

### 2.5. Analyses

For descriptive statistics and summary tables, chronic pain (CP) was defined as pain severity (NPS) of greater than or equal to 4 at the second time point. The continuous underlying indices of NPS and pain interference were used in other analyses.

Demographic and clinical characteristics were summarized using counts with percentages for categorical variables and mean with standard deviation and range for continuous variables. These were compared between groups using the chi-squared test and Mann-Whitney test, respectively. The comparisons of CP between individual injury mechanisms used Fisher’s exact tests with permutation-based adjustment over the possible mechanisms to control the overall type I error rate.

Based on initial bivariate exploratory analyses, the biomarkers (2-AG, AEA, cortisol) were log-transformed to improve the linearity of the relationships and reduce skewness. Pearson’s correlation coefficient was used to quantify the strength of association between different biomarkers and between biomarkers and pain indices as continuous variables. The *p*-values were adjusted for multiple comparisons using the Benjamini–Hochberg method that controls the false discovery rate. For these exploratory analyses, FDR < 0.1 was considered a significant correlation.

A path model was developed to analyze the relationships among the circulating eCB and cortisol concentrations and pain measures at both time points, adjusting for covariates. The initial model structure was constructed based on biological plausibility, measurement timing, and the results of our exploratory correlational data. Specifically, the following variable groups were considered: sex and injury severity score (ISS); hospital and follow-up 2-AG and cortisol concentrations; self-reported pain score at hospitalization; and pain severity and interference scores obtained from the BPI at follow-up. In the initial model, sex and ISS were allowed to influence both hospitalization and follow-up pain measures. In addition, concentrations of 2-AG and cortisol were assumed to be correlated and could affect pain and biomarker measures at either time point. No direct effect of pain on 2-AG or cortisol was included. Hospitalization pain scores were included as a predictor of the follow-up pain measures. The effects of sex, cortisol, and 2-AG on pain were constrained to have equal strength at both hospitalization and follow-up, and the correlation between 2-AG and cortisol was also constrained to be the same at both time points. The model was fitted using the full information maximum likelihood method, which is a maximum-likelihood-based method that can incorporate missing-at-random observations [39].

The initial model was then simplified to find a more parsimonious description. Paths with non-significant effects with standardized coefficients under 0.1 in absolute value were removed, monitoring that goodness of fit indices continue to fall in their acceptable ranges and prioritizing models with lower Bayesian Information Criterion (BIC). Table 1 shows the goodness of fit indices of the initial and reduced models, indicating an excellent fit for both.

All analyses were performed using SAS 9.4 (SAS Institute, Cary, NC, USA) using the CALIS procedure for the path analysis. Unless otherwise noted, a two-sided 5% significance level was used.

## 3. Results

### 3.1. Demographic Results

Demographic information, pain data, and biospecimens were collected from hospitalized participants an average of 2.5 days following injury (range 1–10 days; hospitalization time point). Pain data and blood samples were also obtained 5–10 months post-injury (average 192 days; range 156–286; follow-up time point). The demographic and clinical information related to the injury for the total population of participants are shown in the second column of Table 2. The sample was predominately male (69.4%) and spanned the entire adult age range. Forty-five percent of the sample self-identified as Black or African American and 7.5% as Hispanic or Latino.

The presence of pain was assessed at the follow-up visit and chronic pain (CP) was defined as an NPS of 4 or greater and was endorsed by 50 individuals (34%). The demographic and clinical characteristics of the no chronic pain (NCP; NPS < 4) and CP subgroups of the sample were determined and compared (Table 2). There were no significant differences between the NCP and CP subgroups with regard to age or sex. There was no difference in the time that elapsed between the injury and follow-up visit between the NCP and CP subgroups. The ISS and NPS scores at the time of hospitalization were both significantly higher in the CP group than in the NCP group. Those not in a committed relationship were more likely to have chronic pain. One hundred forty-five of the 147 subjects in the study were treated with opiate analgesics at the time of injury, so the impact of opiates on chronic pain could not be studied in this cohort. Individuals were asked about cannabis use at both the time of hospitalization and at the follow-up assessment; there was no difference in reported use between the NCP and CP subgroups (data not shown).

### 3.2. Mechanisms of Injury

The mechanisms of traumatic injury were examined in the entire sample and compared between the NCP and CP subgroups. The three most prevalent mechanisms of injury were motor vehicle crashes (32%), falls (17%), and gunshot wounds (16%). A significantly greater proportion of those in the CP than the NCP subgroup was injured by gunshot wounds (28%, *p* = 0.023).

### 3.3. Correlational Analyses of Biomarkers and Pain

Correlational analyses were used to test our initial hypotheses that circulating concentrations of the eCBs and cortisol are correlated with acute and chronic pain (Table 2). For these exploratory analyses, False Discovery Rate (FDR) <0.1 was considered a significant correlation. At the time of hospitalization, cortisol concentrations were significantly, negatively correlated with acute pain; neither 2-AG nor AEA concentrations were correlated with acute pain. There was a modest, positive correlation between cortisol and 2-AG at hospitalization.

At follow-up, none of the biomarkers were correlated with pain measures. However, a significant, positive relationship between 2-AG and cortisol occurred while the concentrations of AEA and cortisol were negatively correlated.

To test the predictive value of the biomarkers, correlational analyses were carried out between biomarker concentrations at the time of hospitalization and indices of chronic pain 5–10 months after injury. 2-AG concentrations at the time of hospitalization were positively correlated with the degree of pain interference with activities of daily living 5–10 months after the injury.

### 3.4. Model Incorporating Biomarkers, Demographics, and Pain Measures

A path model was developed to analyze the relationships among the biomarkers, pain measures, and covariates (Figure 1). Table 1 shows the goodness of fit indices of the initial and reduced models, indicating an excellent fit for both.

The strongest relationships identified using this model were: (1) a positive relationship between serum concentrations of 2-AG at hospitalization and pain severity at follow-up; and (2) a negative relationship between cortisol and pain severity at both time points. Both 2-AG and cortisol concentrations at hospitalization were positively correlated with 2-AG concentrations at follow-up, and cortisol concentrations at hospitalization and follow-up were positively correlated with each other. ISS influenced pain severity at follow-up directly and indirectly through a relationship with 2-AG concentrations at hospitalization. The female sex exerted a significant, positive effect on pain severity at both time points, independent of other factors in the model. Cortisol at hospitalization affected cortisol at follow-up, and pain severity affected interference with activities of daily living at follow-up.

## 4. Discussion

While pain after injury is an expected outcome, continued, unresolved pain is a troubling consequence of traumatic injury. This study explored relationships among biomarkers of stress and pain severity measured at two-time points (hospitalization and five to ten months post-injury) in a sample of 147 traumatically injured subjects to explore biological relationships underlying the transition from acute to chronic pain. Biomarkers, clinical measures, and pain scores were determined within days of the traumatic injury and again five to ten months later-a time period when physical injuries are healed. Moderate to severe chronic pain was present in 34% of subjects 5–10 months after injury, which is a lower percentage than found in a previous study of an injured patient population in a community-based sample [3]. In that study, which was carried out four months after the injury, 43% of patients had moderate to severe pain, 50% had moderate to severe life interference associated with the development of chronic pain, and 50% continued to use opioids to treat their chronic pain four months after a traumatic injury [3].

In the current study, those without a committed relationship were significantly more likely to have chronic pain. This is aligned with the results of a recently published large study (>900 participants) which found that those with limited social support were significantly more likely to exhibit chronic pain, functional limitations, and poor mental health outcomes following moderate-severe traumatic injury [40].

In our study, those injured due to a gunshot wound were more likely to develop chronic pain. Given that civilian gunshot injuries are associated with interpersonal violence, these data support the role of distress as a risk factor for developing chronic pain [41].

As has been reported previously [42], the path analysis demonstrated a significant, positive effect of the female sex on both acute and chronic pain. Likely because our sample was only about one-third women, this difference did not reach significance in univariate analyses, but it did go in the same direction as the path analysis. This difference was most likely because the path analysis had more power with pain scores as a continuous variable and the path model explained some of the variability in the pain scores reducing the unexplained variability and making it easier to detect other effects.

Numerical pain scores at the time of injury were positively associated with the severity of chronic pain. This is in accord with the well-accepted notion that acute pain causes changes in the sensory pathway and pain-related brain circuits resulting in sensitization and chronic pain [43]. Both univariate and path analyses also identified a significant positive correlation between the severity of the physical injury (measured using the ISS) and acute pain at the time of injury and path analysis revealed a moderate but significant positive association between ISS and chronic pain. This result differed from findings in a previous study by our group that did not find ISS to correlate with the development of chronic pain [44], which suggests that ISS has a weak influence on chronic pain. Interestingly, the path model did not identify a significant association between ISS and acute pain, suggesting that ISS contributes to the risk for chronic pain beyond solely an enhancement of acute pain.

Cortisol concentrations were significantly negatively correlated with contemporaneous pain measures both in the univariate analyses and in the path analysis. For the path analysis, we made the assumption that the relationship between pain and cortisol would be independent of sampling time, reasoning that the biological relationship between them would not be altered by time since injury. A rerun of the model without this assumption did not appreciably change the relationships between cortisol and pain at either time, suggesting that this is a valid assumption.

While acute pain tends to increase concentrations of cortisol in healthy individuals, we found a significant, negative relationship between contemporaneous measures of circulating cortisol and pain both in the days after the traumatic injury and at 5–10 months follow-up. The expectation of high cortisol during acute pain was not seen in this sample. There are several possible explanations; the first is the timing of the cortisol measurements, which were several days on average after the injury. It is possible that cortisol was depleted at this stage due to the significant stress of the injury itself. Interestingly, previous studies have found that Blacks and Whites differ in cortisol responses to pain and stress, with Blacks exhibiting a lower response than Whites [45]. Given that our study population was enriched in individuals from racial and ethnic minorities, it is possible that the demographics of our sample contributed to this relationship. On the other hand, hypocortisolism has been linked to chronic pain disorders, including fibromyalgia, chronic fatigue syndrome, chronic pelvic pain, and temporomandibular disorder [46,47]. As suggested by Hannibal and Bishop, hypocortisolemia can potentiate and prolong chronic pain due to increased inflammation, which can increase pain and increase the risk for depressed mood, an additional risk factor for chronic pain [48].

The lack of association between levels of eCBs at hospitalization and acute pain refutes our hypothesis that pain measures correlate with eCBs at baseline. However, the positive correlation between cortisol and 2-AG at the time of hospitalization supports our hypothesis and is in accord with preclinical data demonstrating that glucocorticoid receptor (GR) activation increases 2-AG synthesis in the brain [49] and periphery [50], although the possibility that cortisol and 2-AG are elevated independently cannot be ruled out. In this regard, ISS and 2-AG were positively associated, suggesting that the severity of the traumatic injury may contribute to 2-AG concentrations independently from cortisol.

Our primary goal in this study was to test the predictive value of the biomarkers measured at the time of injury for the development of chronic pain. Bivariate analyses indicated that 2-AG concentrations at the time of hospitalization were positively associated with the degree of pain interference with activities of daily living measured using the Brief Pain Inventory at follow-up. The path analysis also identified a significant and positive association between circulating 2-AG concentrations at the time of injury and pain severity at the follow-up visit and a nearly one-to-one correspondence between pain severity and pain interference. These data support our hypothesis that circulating concentrations of 2-AG have value as a biomarker for the risk of developing chronic pain.

On the other hand, neither analysis demonstrated significant associations between measures of pain severity or interference at follow-up and circulating concentrations of 2-AG measured at the same time. This finding contrasts with studies in which circulating 2-AG was found to be higher compared to pain-free control groups in individuals with several types of chronic pain, including fibromyalgia, irritable bowel syndrome, and neuropathic pain [17,18,51]. Differences in the duration and type of chronic pain could underlie the difference in findings.

It is yet unknown if elevated 2-AG concentrations during the time of injury *per se* are mechanistically involved in the severity of pain months later. Given the preclinical data that CB1R activation is associated with reduced pain in many models and that highly elevated 2-AG concentrations result in reduced CB1R density [52], it is possible that CB1R signaling is down-regulated by the high 2-AG concentrations that occur following injury. This could result in increased pain perception at the time of injury, a known risk factor for the development of persistent pain [53]. Our finding of a significant relationship between pain severity at hospitalization and follow-up supports this notion, as do the extremely high concentrations of 2-AG during the peritraumatic period. However, pain severity was not related to circulating 2-AG concentrations at hospitalization, which would be expected if this were the mechanism. An alternative hypothesis, based upon the findings that chronic pain is accompanied by widespread changes in brain circuits [43] and 2-AG/CB1 signaling affects synaptic activity throughout the brain [54], is that excessive 2-AG-mediated signaling at the time of injury contributes to lasting changes in circuits that subserve chronic pain. Further studies are needed to explore these and other possible mechanisms.

There were no interactions between circulating concentrations of the second endocannabinoid, AEA, and the other measures in this study. While 2-AG and AEA are both endogenous ligands of the cannabinoid receptors, there are differences in the triggers for their mobilization and their reported associations with psychological indicators in humans [55]. Previous studies in humans have found that AEA concentrations are more likely to be associated with anxiety while 2-AG is more likely to be associated with depression [56,57]. Interestingly, 2-AG concentrations at hospitalization were also associated with increased risk for depression at follow-up [32], while AEA concentrations at hospitalization were associated with risk for the development of chronic PTSD [33] in the participants of this study.

This study was not without limitations. First, we measured the endocannabinoids at only two time points (hospitalization and at least 5 months later), which provided partial longitudinal data over a period of time when the participants were recovering from their injury and were undergoing important changes in symptoms. However, we do not know the trajectory, and importantly, we do not know when the elevated 2-AG concentrations returned to normal values. Our hypothesis that excessive elevation of eCB signaling suggests that individuals with a prolonged increase in 2-AG would be at greater risk for chronic pain. Measuring the eCBs over multiple time points would allow us to test this hypothesis. In addition, the follow-up visits occurred over a broad range of times (5.2 months to nearly 10 months), which could add significant variability to the results. The blood collection times were dictated by the situation of the participant, which resulted in a fairly wide range of elapsed time since injury and did not differ between the NCP and CP groups. Similarly, the time of day for the blood collections was not controlled for, which is a limitation because of the circulating concentrations of both cortisol and 2-AG exhibit strong circadian rhythms [58]. Moreover, the literature has noted that the microbiota can affect the ECSS [59,60,61]. However, this was not evaluated within our study. Finally, we did not screen participants for pre-existing chronic pain, which could be a possible confound for assessing chronic pain associated with the current traumatic injury. However, the participants were asked at the follow-up to evaluate the severity of their pain related to the injury a few months earlier.

## 5. Conclusions

In conclusion, the results of this preliminary study suggest that both 2-AG and cortisol concentrations are associated with the risk for development of chronic pain following injury and add the endocannabinoid system to the list of stress-responsive systems that are associated with long-term consequences of an injury. We have previously published that 2-AG concentrations at the time of injury are also positively associated with risk for the development of major depression 6–9 months later [32]. Together, these studies indicate that 2-AG concentrations, which are roughly six times higher following trauma than under normal conditions, could serve as a general biomarker for risk for negative psychological states following traumatic injury. Future studies will probe the more interesting possibility that strong engagement of endocannabinoid signaling in the periphery and brain occurs in the aftermath of significant physiological stress and while this response may be beneficial at the time of injury, it could result in long-term negative effects in certain individuals.

## Figures and Tables

**Figure 1 biomedicines-10-01599-f001:**
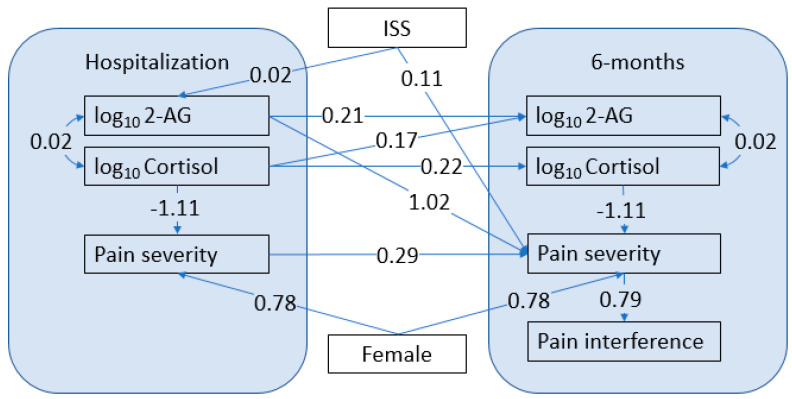
Model of the Relationship of Traumatic Pain, Cortisol, and Endocannabinoid Levels. The numbers on the arrows are regression coefficients, representing the effect of a 1-unit increase in the predictor on the outcome. For the log-transformed biomarkers, a 1 unit increase on the log scale corresponds to a 10-fold increase in the original value. All coefficients shown are significant with *p* < 0.05.

**Table 1 biomedicines-10-01599-t001:** Goodness of fit indices of the initial and reduced models.

Goodness-of-Fit Measure	Guideline for Acceptable Fit	Initial Model	Reduced Model
Χ^2^/df	<3	14.9/11 = 1.3	20.9/23 = 0.91
Standardized root mean square residual (SRMR)	<0.080	0.041	0.049
Root mean square error of approximation (RMSEA) with 90% confidence interval	<0.06, upper limit < 0.08	0.05 (0.0–0.11)	0.0 (0.0–0.06)
Comparative fit index (CFI)	>0.95	0.98	1.0
Bayesian information criterion (BIC)	Lower value implies more parsimonious fit	228.6	175.9

**Table 2 biomedicines-10-01599-t002:** Demographic and clinical data of the study participants.

Parameter	Total Sample	NCP (NPS Score < 4)	CP (NPS Score ≥ 4)	*p*-Value (NCP Compared to CP)
N	147	97	50 (34%)	
Mean Age (SD, range)	42.5 (16.4, 18–89)	42.2 (17.5, 18–89)	42.9 (14.0, 20–74)	*p* > 0.1
Sex				*p* > 0.1
Female (percent)	45 (30.6)	28 (28.9)	17 (34.0)	
Male (percent)	102 (69.4)	69 (71.1)	33 (66.0)	
Race/Ethnicity				0.09
Non-Hispanic White	68 (46.3)	52 (53.6)	16 (32.0)	
Black or African American	66 (44.9)	37 (38.1)	29 (58.0)	
Hispanic or Latino	11 (7.5)	7 (7.2)	4 (8.0)	
Native American/Alaskan Native	2 (1.4)	1 (1)	1 (2)	
Highest Educational Level Completed				0.09
Advanced degree (master’s or higher)	10 (6.8)	10 (10.3)	0 (0.0)	
College graduate	24 (16.3)	15 (15.5)	9 (18.0)	
Graduated high school, some college	52 (35.4)	37 (38.1)	15 (30.0)	
High school graduate, no college	36 (24.4)	20 (20.6)	16 (32.0)	
Less than high school	25 (17.0)	15 (15.5)	10 (20.0)	
In a committed relationship				0.056
No	57 (39.3)	32 (33.7)	25 (50.0)	
Yes	88 (60.7)	63 (66.3)	25 (50.0)	
Time between injury and follow-up assessment for chronic pain and blood draw (SD, range)	192 days (22, 156–286)	191 (19, 156–240)	194 (26, 160–286)	*p* > 0.1
Injury severity score (ISS; SD, range)	10.1 (5.9, 0–29)	9.1 (5.2, 0–24)	12.2 (6.6, 0–29)	0.002
Numerical pain score at hospitalization (SD, range)	5.8 (2.4, 0–10)	5.2 (2.3, 0–10)	6.9 (2.3, 1–10)	<0.001

NCP: no chronic pain and CP: chronic pain subgroups determined at follow-up. Age, time between injury and follow-up, severity scores (ISS), and acute pain scores were compared between the NCP and CP groups using Mann–Whitney non-parametric *t*-tests; other comparisons were made using the Chi-squared test.

## Data Availability

Not applicable.

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
