# Peer review of "Serum Concentrations of the Endocannabinoid, 2-Arachidonoylglycerol, in the Peri-Trauma Period Are Positively Associated with Chronic Pain Months Later"

_biomedicines, 2022, doi:10.3390/biomedicines10071599_

Round 1

Reviewer 1 Report

The interpretation of the paper relies on a misconception of the the endocannabinoid system. 2-AG is only one and is often oppositely regulated compared with anandamide and other ethanolamide endocannabinoids (PEA; OEA). Without analysis of at least AEA and 2AG one cannot conclude that an upregulation of the endocannabinoid system causes pain in the trauma situation. 2-AG is pro-inflammatory in most circumstances and a major source in the periphery in fat tissue (hence BMI is crucial) and 2-AG levels are different in males and females. Unfortunately, none of the data is presented. There are only demographic tables and depiction of a model but no actual data such as scatter plots showing concentrations, XY plots to reveal associations of two parameters, multiple line plots to reveal time courses of individual patients. In addition, the authors measured 2-AG in serum (instead of plasma), meaning that the concentrations they find originate to a great extent from platelets which release 2-AG on clotting. Serum levels of 2-AG are not representative for eCB in the brain. It is even more irritating that they used EDTA tubes to obtain serum, which is not possible. They report that they have analyzed AEA and 2-AG but none of the results are presented.

Author Response

Thank you for your review of our study. We agree there are many limitations in this study. We did measure anandamide, OEA and PEA, but found no relationship with outcome variables. Therefore, this data was left out. In this heterogeneous traumatically injured population, many thins cannot be controlled for due to the nature of acute injury and different types of injury. However, this is the first paper of its kind to see a relationship between traumatic pain and the ECSS and cortisol, two stress responsive systems. Therefore, we believe it is an important finding that can direct future studies that can address our limitations. 

Reviewer 2 Report

This study nicely illustrates the relation between the 2-arachidonolyglycerol, cortisol concentrations, and the development of chronic pain after an injury. The study highlights and paves the way for potential future studies on the role of endocannabinoid signaling and chronic pain. No further comments

Author Response

Thank you very much for your review of our manuscript. 

Reviewer 3 Report

The study proposed by the authors concerns endocannabinoid signaling and the hypothalamus-pituitary-adrenal axis with the involvement of acute to chronic pain.
The manuscript is well organized, very clear and even the rationale is easy to interpret.
Perhaps these endocannabinoid manuscripts might help in the introduction:

Freitas MA, Vasconcelos A, Gonçalves ECD, Ferrarini EG, Vieira GB, Cicia D, Cola M, Capasso R, Dutra RC. Involvement of Opioid System and TRPM8 / TRPA1 Channels in the Antinociceptive Effect of <i> Spirulina platensis </i>. Biomolecules. 2021 Apr 17; 11 (4): 592.

 Vieira G, Cavalli J, Gonçalves ECD, Braga SFP, Ferreira RS, Santos ARS, Cola M, Raposo NRB, Capasso R, Dutra RC. Antidepressant-Like Effect of Terpineol in an Inflammatory Model of Depression: Involvement of the Cannabinoid System and
D2 Dopamine Receptor. Biomolecules. 2020 May 20; 10 (5): 792. .

Gonçalves ECD, Baldasso GM, Bicca MA, Paes RS, Capasso R, Dutra RC. Terpenoids, Cannabimimetic Ligands, beyond the <i> Cannabis </i> Plant. Molecules. 2020 Mar 29; 25 (7): 1567.

I suggest the authors to add a graphical abstract, in this way the results are more immediate.
Do the authors think the microbiota can affect?

Author Response

Thank you for your review of our manuscript. I have added the suggested research articles to the manuscript and a statement that the microbiota could affect the ECSS but it was not evaluated within this study. 

Round 2

Reviewer 1 Report

The authors have not addressed my concerns. The manuscript is unaltered. "Red-top tubes" (as they write now) are not serum collecting tubes. It is still unclear if eCBs were analysed in serum or in plasma which is a huge difference. Apparently, the authors do not know the difference. The pre-analytical procedures are crucial in eCB analysis. If EAE, OEA and PEA were analyzed, the results should be shown as a suppl. file.